# A homogeneous SIRPα-CD47 cell-based, ligand-binding assay: Utility for small molecule drug development in immuno-oncology

Teresa L. Burgess[1], Joshua D. Amason[1,2¤a], Jeffrey S. Rubin[1], Damien Y. Duveau[3], Laurence Lamy[3], David D. Roberts[2], Catherine L. Farrell[1], James Inglese[3], Craig J. Thomas[3], Thomas W. Miller[1,2¤b]*

1 Paradigm Shift Therapeutics LLC, Rockville, Maryland, United States of America, 2 Laboratory of Pathology, Center for Cancer Research, National Cancer Institute, National Institutes of Health, Bethesda, Maryland, United States of America, 3 Division of Preclinical Innovation, National Center for Advancing Translational Studies, National Institutes of Health, Rockville, Maryland, United States of America

¤a Current address: Department of Ophthalmology, Duke University School of Medicine, Durham, North Carolina, United States of America.
¤b Current address: Institut Paoli-Calmettes, Cancer Research Center Marseille, Marseille, France
* thomas.miller@pstherx.com

**Data Availability Statement:** Data supporting assay optimization and validation are available at Zenodo (DOI: 10.5281/zenodo.3711485). The

## Abstract

CD47 is an immune checkpoint protein that downregulates both the innate and adaptive anti-tumor immune response via its counter receptor SIRPα. Biologics, including humanized CD47 monoclonal antibodies and decoy SIRPα receptors, that block the SIRPα-CD47 interaction, are currently being developed as cancer immunotherapy agents. However, adverse side effects and limited penetration of tumor tissue associated with their structure and large size may impede their clinical application. We recently developed a quantitative high throughput screening assay platform to identify small molecules that disrupt the binding of SIRPα and CD47 as an alternative approach to these protein-based therapeutics. Here, we report on the development and optimization of a cell-based binding assay to validate active small molecules from our biochemical screening effort. This assay has a low volume, high capacity homogenous format that relies on laser scanning cytometry (LSC) and associated techniques to enhance signal to noise measurement of cell surface binding. The LSC assay is specific, concentration dependent, and validated for the two major human SIRPα variants (V1 and V2), with results that parallel those of our biochemical data as well as published studies. We also utilized the LSC assay to confirm published studies showing that the inhibition of amino-terminal pyroglutamate formation on CD47 using the glutaminyl cyclase inhibitor SEN177 disrupts SIRPα binding. The SIRPα-CD47 interaction could be quantitatively measured in live and fixed tumor cells. Use of fixed cells reduces the burden of cell maintenance and provides stable cell standards to control for inter- and intra-assay variations. We also demonstrate the utility of the assay to characterize the activity of the first reported small molecule antagonists of the SIRPα-CD47 interaction. This assay will support the screening of thousands of compounds to identify or validate active small molecules as hits, develop structure activity relationships and assist in the optimization of hits to leads by a typical iterative medicinal chemistry campaign.

synthetic chemistry data are contained in the Supporting Information file. The screening data generated in this study has been deposited in PubChem (https://pubchem.ncbi.nlm.nih.gov/classification/#hid=1), use "keyword" = AID in the pulldown menu. The CD47-SIRPα protein-protein interaction—AlphaScreen assay qHTS validation PubChem AID is 1347059. The CD47-SIRPα protein-protein interaction—LANCE TR-FRET assay qHTS validation PubChem AID is1347057.

**Funding:** This work was supported in part by the Avon Foundation for Women (02-2015-045, 02-2014-051, 02-2015-095, 02-2017-034; TWM, TLB, CLF), the National Cancer Institute (1U01CA218259-01A1; TWM, TLB, CLF, JSR) the Intramural Research Programs of NCATS (Project 1ZIATR000053-03, JI) and NCI (Project ZIASC009172, DDR). The funder provided support in the form of salaries for authors [TWM, JDA, TLB, CLF], but did not have any additional role in the study design, data collection and analysis, decision to publish, or preparation of the manuscript. The specific roles of these authors are articulated in the 'author contributions' section.

**Competing interests:** TWM, TLB, and CLF are co-owners of Paradigm Shift Therapeutics, a company focused on the development of CD47-targeting therapeutics for immune-oncology. JDA was an employee of, and JSR is a scientific advisor to, Paradigm Shift Therapeutics. This does not alter our adherence to PLOS ONE policies on sharing data and materials.

# Introduction

Cancer arises in part when tumor cells acquire mechanisms to disrupt both innate and adaptive immunity to evade immune surveillance [1–3]. Immune checkpoint inhibitors are being developed as a therapeutic strategy to enable the immune system to eradicate neoplasia, especially disseminated tumor cells [4,5]. Antibodies to inhibit the adaptive immune checkpoints PD-1/PD-L1 and CTLA-4/(CD80, CD86) have proven to be remarkably efficacious in a subset of patients [6–10]. Chimeric T-cell receptors and dendritic cell vaccines are also promising treatment modalities to boost the adaptive immune response [11,12]. Another emerging strategy focuses on enhancing innate tumor immunity by targeting the SIRPα-CD47 axis [13].

CD47 is widely expressed on cells and binds to its counter-receptor SIRPα, which is expressed on the surface of macrophages and antigen-presenting cells (APCs), to inhibit phagocytosis and antigen presentation [14–18]. This is a basic mechanism of innate immune tolerance—the so called 'don't eat me" signal. Increased expression of CD47 by tumor cells inhibits their phagocytosis, a crucial way in which they evade immune surveillance [19]. Many preclinical studies have shown that abrogation of the SIRPα-CD47 interaction, especially when combined with tumor targeting antibodies or chemo/radiotherapy, promotes cancer cell death and improves survival [19–27]. Various biologic agents targeting the SIRPα-CD47 axis, including monoclonal antibodies and decoy receptors, are in early clinical development as cancer immunotherapies[28–31]. Encouraging results for one of these agents were recently observed in a phase 1b clinical trial [32].

We have initiated a novel strategy to disrupt the SIRPα-CD47 protein-protein interaction (PPI) that is focused on drug-like small molecules (SMs)[33]. In contrast to the large biologics, SM inhibitors can be designed to specifically block the binding of CD47 to SIRPα without interfering with its other binding partners, e.g. members of the thrombospondin and integrin families[34]. This strategy will allow the SMs to serve as specific molecular probes of SIRPα-CD47 signaling in experimental models. Moreover, along with the pharmacodynamic advantages and potential for oral delivery, such specificity may favor their use as therapeutics by reducing adverse side effects. Recently, we developed a set of quantitative high throughput screening (qHTS) assays and identified SMs that inhibit the SIRPα-CD47 interaction[33].

In the present report, we describe a sensitive, high capacity, cell-based, SIRPα-CD47 binding assay with characteristics that will enhance the identification of preclinical and clinical agents. It combines laser scanning cytometry (LSC)[35] with a 384-well or 1536-well plate format and the addition of reagents without intervening washing steps (homogeneous format). This maximizes the interaction of reagents in a small volume, minimizes the consumption of materials and permits the direct comparison of many compounds in broad concentration-response titrations. Additionally, the homogeneous format results in a shorter duration assay (~45 min.) more conducive to the use of live cells. The use of formalin-fixed as an alternative to live cells adds to the convenience of the assay and potential applications. We used this assay to further validate a number of active SMs from our qHTS program, and quantitatively characterize the impact of the pyro-GLU N-terminal post-translational modification of CD47 on SIRPα binding described by others[36].

# Methods

## Reagents

SIRPα (with and without biotin conjugation) and CD47 were produced as described in [33]. Anti-CD47 clone B6H12-Alexa488 was purchased from Santa Cruz, clone B6H12 (non-conjugated) was purchased from Ebioscience, clone CC2C6-FITC and clone CC2C6 (non-

conjugated) were purchased from Biolegend. Streptavidin-Alexa488, Neutravidin-Alexa488 and DRAQ5 nuclear stain were purchased from ThermoFisher. PBS buffer was purchased from Gibco. HEPES buffer contains 10 mM HEPES pH 7.5, 0.15 M NaCl, all of which were purchased from Sigma.

**Cell lines and tissue culture.** Jurkat T lymphoma cells and A2058 melanoma cells were obtained from American Type Culture Collection. The CD47(-) Jurkat somatic mutant JinB8 was from Dr. Eric Brown (PMID:10330276). Cells were cultured in RPMI 1640 medium (Gibco, ThermoFisher) supplemented with 10% fetal bovine serum (Gibco, ThermoFisher), 2 mM Glutamine, 25 mM HEPES, and 1 mM sodium pyruvate (Gibco, ThermoFisher). All cells were cultured at 37°C in a humidified incubator under 5% $CO_2$. All Cell lines were Mycoplasma-free as determined with the MycoAlert mycoplasma detection kit (Lonza, Walkersville, MD, USA) and were carried no more than 20 passages from the validated stocks.

**Flow cytometry assay.** Approximately 100,000 cells were harvested, washed with PBS, resuspended in 200 μL of PBS containing 1% human serum albumin and 0.01% sodium azide (FACS Buffer, BD), and added to a 96-well round bottom plate for staining. SIRPα-biotin was added as indicated in Fig 1 (0.1, 0.3 1.0 μM) and incubated for 30 min at room temperature. Cells were then washed by centrifuging at 1000 x g for 5 min, supernatant aspirated, and resuspended in 200 μL of FACS Buffer. Streptavidin-Alexa488 (SAV488) was then added at 0.5 μg/mL for 30 min at room temperature after which the cells were washed again as above. For non-biotin SIRPα competition studies, the non-biotin and biotin conjugated forms were incubated with the cells at the same time at the indicated concentrations. For Anti-CD47 antibody labeling studies, fluorochrome conjugated antibodies B6H12 and CC2C6 were used at

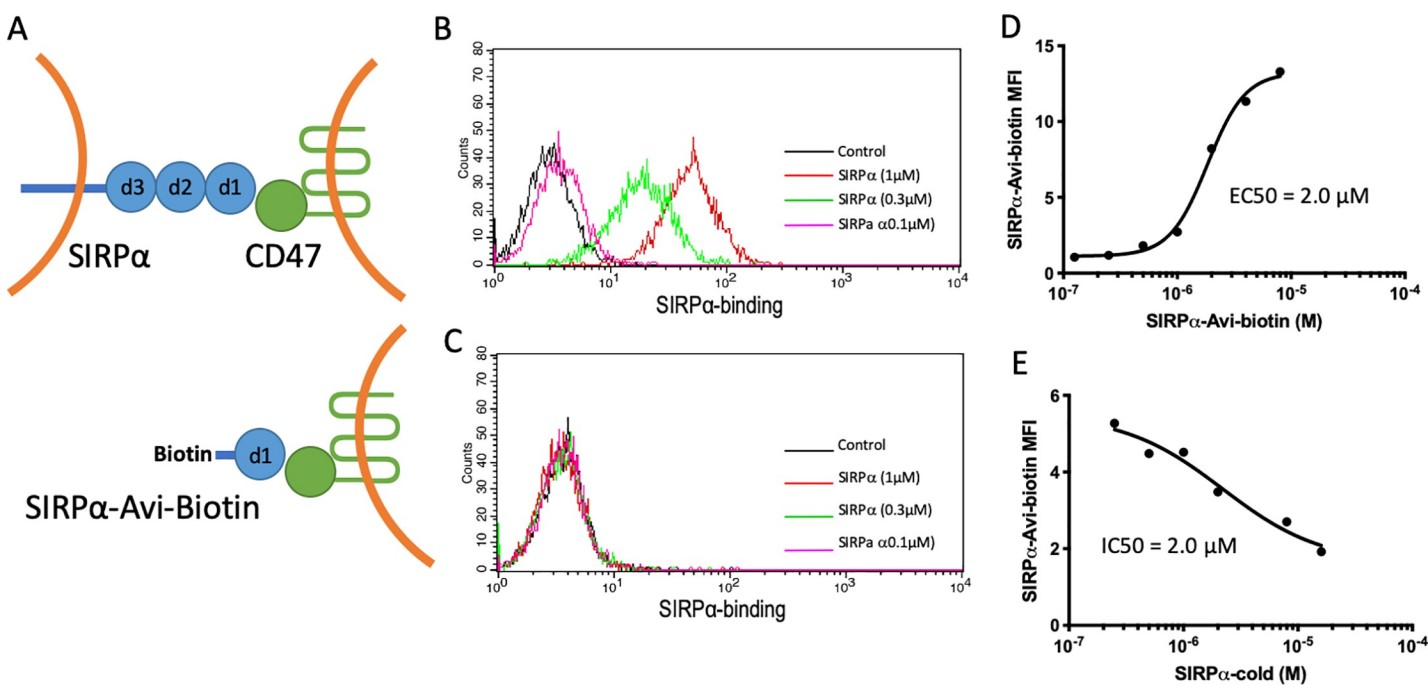

**Fig 1.** SIRPα specifically binds to CD47 on Jurkat cells (A) Schematic of the SIRPα extracellular domains (d1, d2, d3) and the soluble, biotin-conjugated, CD47-interacting SIRPα (d1) construct used in these studies (SIRPα-Avi-biotin). Flow cytometry histograms showing the concentration-dependent binding of SIRPα to (B) CD47(+) or (C) CD47(-) Jurkat cells imaged using Alexa-488 conjugated streptavidin. Control indicates cells treated with SAV-488 alone (no SIRPα). (D) Concentration response plot showing the calculated EC50 of SIRPα binding to CD47(+) Jurkat cells. (E) Concentration-dependent inhibition of SIRPα binding to CD47(+) Jurkat cells by SIRPα-cold (no biotin).

**Table 1. Optimized LSC assay protocol.**

| Sequence | Parameter | Value | Description |
|---|---|---|---|
| 1 | Cells | 2x10$^6$ | Remove 2x10$^6$ Jurkat cells and wash 1x with HEPES buffer and resuspend with 8 mL of HEPES buffer into a 15 mL Falcon tube |
| 2 | Reagent | 3 μL | Add 3 μL of DRAQ5 to cell tube (2 μM final) |
| 3 | Incubate | 5 min | Incubate cell tube in incubator for 5 minutes (37C, 95%RH, 1atm, 5%CO$_2$) |
| 4 | Plate | 20 μL | Add 20 μL/well of cells (5000 cells/well) with multichannel pipet or Multidrop dispenser to Greiner 384 well plate |
| 5 | Reagent | 10 μL | Add SIRPα-biotin,10 μL/well |
| 6 | Centrifuge | 30 sec | Centrifuge plate for 30 sec at 1000 RPM |
| 7 | Incubation | 30 min | Incubate plates at room temperature for 30 min, protected from light |
| 8 | Reagent | 10 μL | Mix 10 μL of NAV488 with 5 mL of Hepes buffer, add 10 μL/well (0.5 μg/mL final) |
| 9 | Centrifuge | 30 sec | Centrifuge plate for 30 sec at 1000 RPM |
| 10 | Incubation | 30 min | Incubate plates at room temperature for 60 min, protected from light |
| 11 | Detector | Mirrorball | Measure signals with Mirrorball (DRAQ5 on FL4 triggering NAV488 on FL2) |
| 12 | Analyze | Prism | Data processing and analysis using GraphPad Prism |

1 μg/mL final concentration. Fluorochrome-labeled isotype control antibodies were used as negative controls for nonspecific immunoglobulin binding. Cells were washed as above, samples were resuspended in 200 μL of FACS Buffer and measured on a FACS CALIBER II and Accuri C6 Flow Cytometer (BD BioSciences). We adhered to the guidelines for the use of flow cytometry and cell sorting in immunological studies.

**Laser scanning cytometry assay.** The assay was conducted as in Table 1, and all manipulation/incubations were subsequently carried out at room temperature. Optimized reagent amounts for 384 and 1536 well plates are as in Table 2. DRAQ5-labeled cells were added to 384 or 1536 well plates using a multichannel pipet or a Multidrop dispenser (ThermoFisher). Competing ligand/antibody or control mixed with SIRPα-biotin were added to the cells using a multichannel pipet or Mosquito liquid handler (TTP Labtech). Following NAV488 addition and incubation, cell plates were imaged using the Mirrorball LSC according to Table 1. For measurements involving fixed cells, the cells were fixed in 4% paraformaldehyde for 30 min on ice, washed and then stained as above.

**Small molecule synthesis and testing.** NCGC00138783 is a compound derived from qHTS testing of the NCATS chemical libraries as described in [33]. Based on the confirmed activity of NCGC00138783 in biochemical assays, a synthetic scheme was devised to create analogs with which to develop a structure activity relationship (see S1 Material).

**SEN177 treatment.** A2058 cells were plated on day 0 in 6 well plates at 300,000 cells per well in fully supplemented RPMI as described above. The cells were allowed to adhere overnight and then were treated with SEN177 (Sigma) with a constant DMSO (vehicle) level of 0.9%. The treatment was repeated 24 hours later. 24 hours after the second SEN177 treatment, the cells were harvested using trypsin and fixed (4% paraformaldehyde for 30 min on ice) before being split into 3 samples for analysis in parallel using SIRPα/LSC, CC2C6/flow cytometry, and B6H12/flow cytometry. Data represent the mean and standard error of 3 biological replicates with 3 technical replicates of each biological replicate.

**Table 2. Optimized LSC assay parameters.**

| Plate type | # of Cells | Total Vol. | NAV488 conc. | Incubation time | Read time |
|---|---|---|---|---|---|
| 384 well—Corning 3712; Greiner 781096 | 5000 | 40 μL | 0.5 μg/mL | 65 min | 12 min |
| 1536 well–Aurora E8 | 2500 | 6 μL | 2.0 μg/mL | 65 min | 45 min |

## Results

### SIRPα binding to Jurkat cells detected by flow cytometry

We previously used SIRPα and CD47 recombinant proteins to develop a set of biochemical assays for qHTS of SM libraries to identify inhibitors of the SIRPα-CD47 interaction[33]. The next step toward identifying SIRPα-CD47 inhibiting SMs was to establish an assay that would enable us to test the ability of lead compounds to inhibit SIRPα binding to CD47 naturally expressed by tumor cells. For this purpose, we developed a model where the affinity for CD47 of a truncated biotin-tagged SIRPα, hereafter referred to as SIRPα, was assessed on Jurkat T cells with [CD47(+)] or without [CD47(-)] CD47 expression by flow cytometry (**Fig 1A**). SIRPα bound CD47(+) cells in a concentration-dependent manner, while no binding to CD47 (-) cells was detected (**Fig 1B and 1C**). The specificity of the interaction was demonstrated by both saturable binding (**Fig 1D**) and competition with a non-biotin conjugated SIRPα derivative (SIRPα-cold; **Fig 1E**).

### Development of a SIRPα-CD47 binding assay using laser scanning cytometry

While flow cytometry is routinely used for cell-based ligand binding assays and immunoassays, it has practical limitations with regard to the sample processing speed, reagent consumption, and the number of samples that can be tested in a single experiment. To overcome these limitations, a cell-based SIRPα-CD47 binding assay ideally would include a low cell/reagent consumption, large sampling density for replicates and high-resolution concentration-response titrations. To satisfy these criteria, we developed a binding assay using a LSC platform (Mirrorball, TTP Labtech). Early LSC-style assays developed for cell-based cytokine and GPRC receptor fluorophore-labeled ligand binding were first demonstrated on early prototype plate-based cytometers [35]. The current LSC platform utilizes confocal optics that enable discrimination of cell-based fluorescence from bulk (unbound) fluorescence, permitting a wash-free, homogeneous protocol [37]. Using the same reagents as in the conventional flow cytometry assay described above, we optimized the LSC assay signal and robustness for plate type, cell number, secondary detection reagent concentration, incubation time, and instrument settings (**Tables 1 and 2**). The staining strategy is as illustrated in the schematic drawing (**Fig 2A**). Representative data show the nuclear counter stain employed for cell detection (DRAQ5; **Fig 2B**) and the Alexa488 channel that represents SIRPα binding via neutravidin-Alexa488 (NAV488; **Fig 2C**). Selective detection of Alexa488 signal in the vicinity of cell nuclei reduces the contribution of unbound reagent fluorescence to background signal (see overlay; **Fig 2D**). Each object is recorded (**Fig 2E**) and subjected to object-level and population-level filtering (**Fig 2F**) for quantification of the final analysis parameters, analogous to routine data processing associated with flow cytometry.

### Evaluation and utility of the optimized LSC assay

CD47(+) Jurkat cells were subjected to a titration of SIRPα concentrations to determine EC50 and specificity in comparison to control Jurkat cells lacking detectable CD47 expression [CD47(-)]. The EC50 value of 16 nM obtained in the optimized LSC assay (**Fig 3A**) is ~100-fold lower (greater affinity) than in the flow cytometry assay (see **Fig 1D**), most likely due to differences in the specific protocols that affect reagent avidity (see Discussion). We next evaluated the binding affinity of CD47 to the most common human SIRPα allelic variants, variants 1 and 2 (SIRPαV1 and V2). SIRPαV2 was two-fold more potent than SIRPαV1 in binding to CD47 (**Fig 3B**), a finding consistent with biochemical studies[38] and now demonstrated here in a cell-based assay.

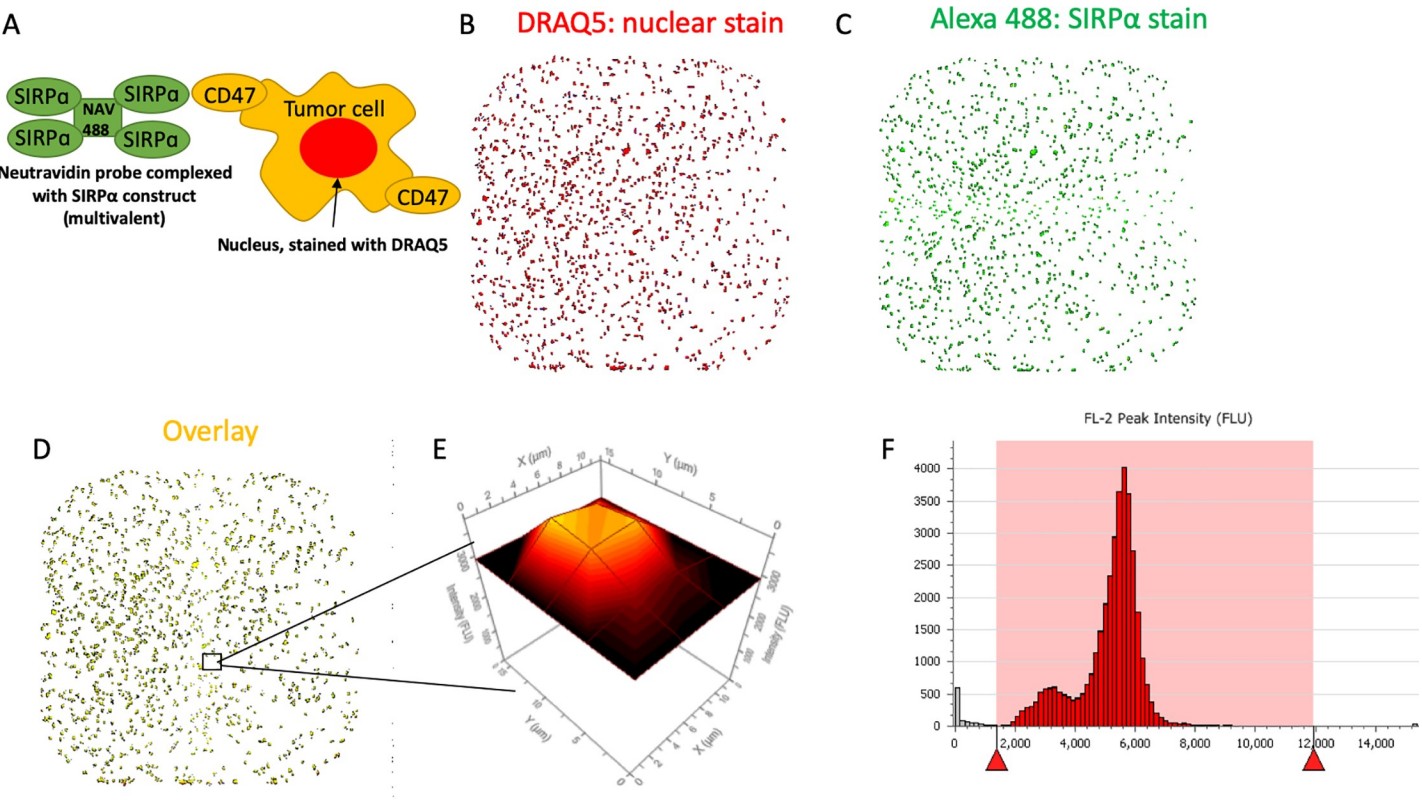

**Fig 2.** (A) Schematic of the LSC assay showing the multivalent Alexa-488 conjugated neutravidin (NAV-488) SIRPα probe binding to CD47+ tumor cells stained with the nuclear dye, DRAQ5. (B-D) Representative images of all CD47+ Jurkat cells in a single well (B) DRAQ5 nuclear stain for object classification (C) Alexa-488 conjugated SIRPα probe signal, and (D) the overlay of B and C indicating cell-specific staining. (E) The size and signal intensity of each object in the well are captured and (F) represented as in a typical flow cytometry histogram. Red triangles and shaded area indicate the signal limits that define the cell population of interest.

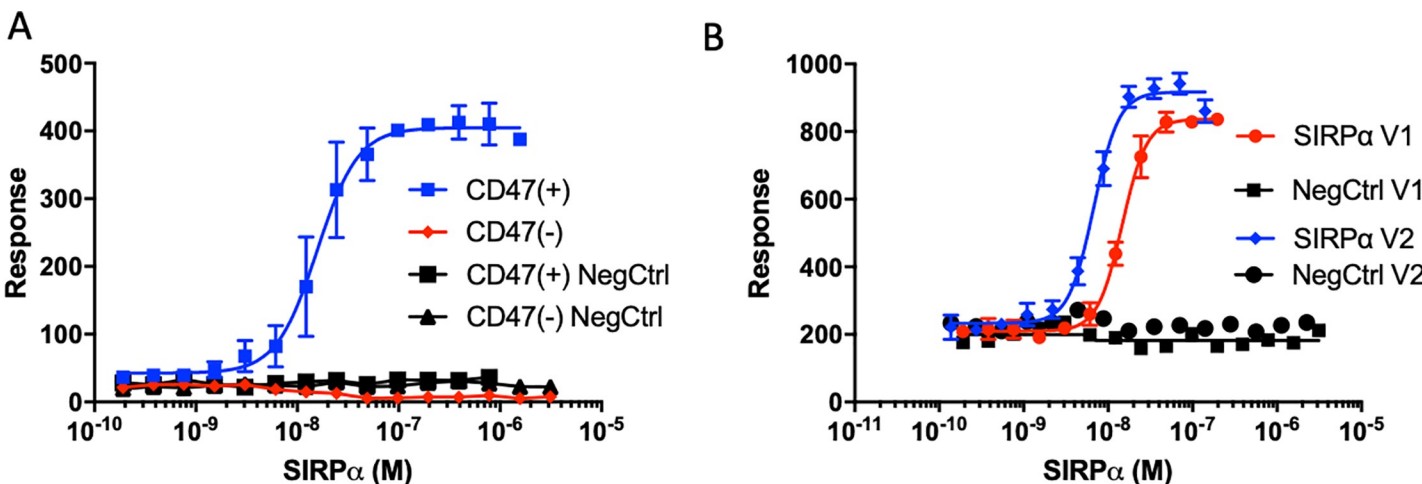

**Fig 3. Quantitative measurements of SIRPα -CD47 binding using the optimized LSC assay platform.** (A) Concentration-dependent SIRPαV2 binding to CD47(+) Jurkat cells using the LSC assay. Negative controls included are CD47(-) Jurkat cells and both CD47(+) and (-) cell lines incubated with NAV-488 and without SIRPα (NegCTRL, nonspecific binding). (B) Comparison of the binding affinities of SIRPαV1 and V2. Negative control indicates background nonspecific binding (-SIRPα). Error bars indicate standard deviation of n = 4 replicates.

To verify the CD47 specificity of the assay, we examined the binding of SIRPα to CD47(+) cells in the presence of a CD47 blocking antibody, clone CC2C6,—an antibody selected for its ability to block both SIRPαV1 and V2 binding to CD47 on CCRF-CEM cells[39]. In these assays, a fixed concentration of SIRPαV1 (20 nM) was co-incubated with the competitor for CD47 binding (CC2C6, B6H12 or recombinant CD47 as indicated). CC2C6 inhibited SIRPαV1 binding to Jurkat cells with an IC50 of 74 pM, comparable to the previously reported CD47 blocking activity of this antibody [40] (**Fig 4A**). Another CD47 blocking antibody, B6H12, also disrupted the SIRPα-CD47 interaction (**Fig 4B**). The IC50 of B6H12 (6 nM) was consistent with the literature[41]. The ability of soluble recombinant CD47 to inhibit SIRPαV1 binding to cell-expressed CD47 was also measured and found to be similar to previously reported biochemical results (0.9 μM vs 0.8 μM, respectively) (**Fig 4B**) [42,43]. Using a similar competition assay, we measured the ability of non-biotin conjugated SIRPαV1 and SIRPαV2 to antagonize the binding of their biotin-conjugated analogs. The IC50s of SIRPαV1 and SIRPαV2 to native, cell surface expressed CD47 were 7.5 and 3.0 μM, respectively (**Fig 4C**). These data were consistent with a previous report that SIRPαV1 is a relatively weaker binder (~2-fold) than SIRPαV2, although the IC50 values we observed were ~5-fold higher (weaker inhibition) than the $k_D$ observed in a pure biochemical assay (surface plasmon resonance spectroscopy, SPR)[38,42]. Taken together, our results established the utility of the LSC platform for the measurement of SIRPα-CD47 binding affinity in live cells.

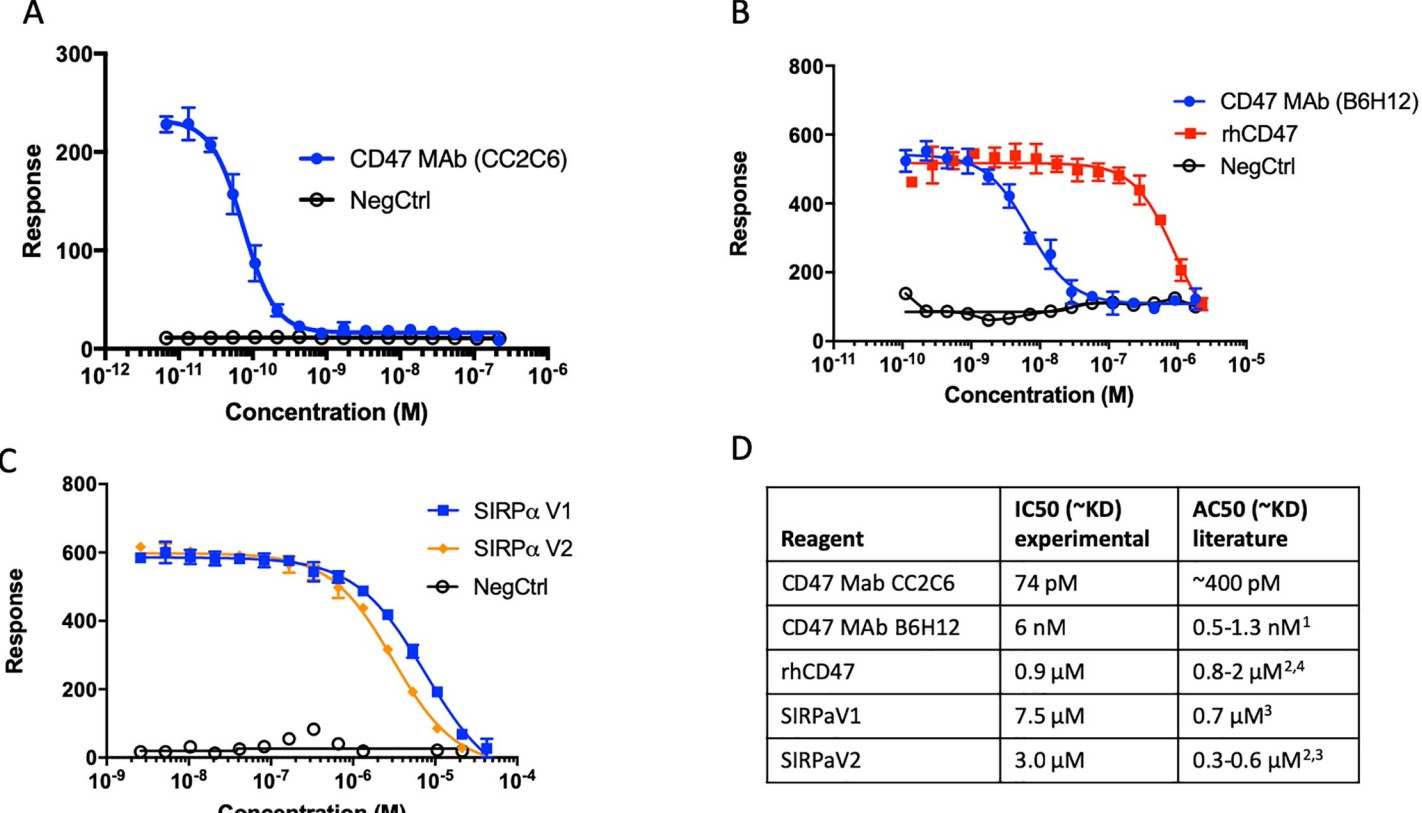

**Fig 4. LSC-based ligand binding activities agree with previous biochemical studies.** SIRPα binding to CD47(+) cells was inhibited using a fixed SIRPα concentration (20 nM) and a titration of the anti-CD47 monoclonal antibody clone (A) CC2C6, (B) B6H12, or recombinant human CD47 (rhCD47) or (C) non-biotin conjugated (cold) SIRPαV1 and V2. Negative control indicates background nonspecific binding (-SIRPα). Error bars indicate standard deviation of n = 4 replicates. (D) The IC50 value for each reagent competing with SIRPα for binding CD47 on Jurkat cells and their comparison to their literature derived biochemical affinities. [1][28]; [2][42]; [3][38]; [4][43].

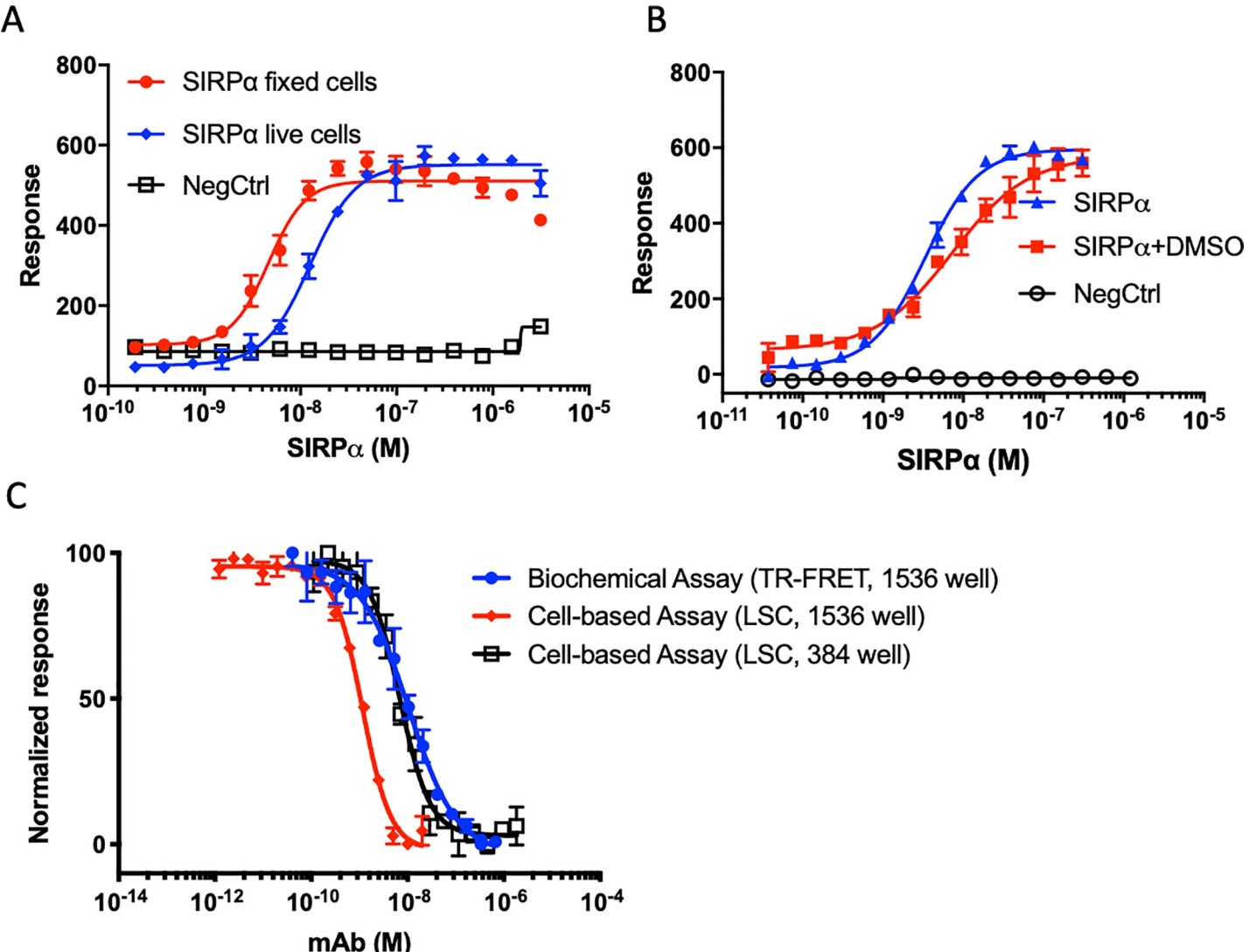

**Fig 5. Adaptation of the LSC assay to measure CD47-SIRPα on fixed vs. live cells.** (A) Comparison of SIRPα binding to CD47(+) Jurkat cells before and after fixation or (B) with and without DMSO. NegCtrl indicates background nonspecific binding (-SIRPα). Error bars indicate standard deviation of n = 4 replicates. (C) Comparison of inhibitor (anti-CD47 Mab clone B6H12) activity in the biochemical assay of CD47-SIRPα (TR-FRET) and the cell-based assay (LSC); using a 384 and 1536 well plate format as indicated. The 1536 well LSC assay employed a fixed SIRPα concentration of 10 nM bound to CD47(+) Jurkat cells. Error bars indicate standard deviation of n = 16 replicates.

### Use of fixed cells in SIRPα-CD47 binding assay with LSC platform

We explored the possibility of further refining the assay to optimize its suitability for a drug development campaign. Because fixed cells would be a stable source of CD47 for automated protocols and inter-day comparisons, we tested the effect of cell fixation on SIRPα-CD47 binding characteristics in our LSC assay (**Fig 5**). SIRPα bound similarly before and after cell fixation, (**Fig 5A**), consistent with a previous report using flow-cytometry [44]. The EC50 was somewhat lower for the fixed cells compared to live cells (1.3 nM vs. 4.6 nM). We also observed that DMSO (1%) had little effect on SIRPα binding in the fixed-cell assay (**Fig 5B**), thus indicating that the assay could be used for testing the inhibitory activity of SMs solubilized in DMSO. Lastly, we compared the activity of a known SIRPα-CD47 binding inhibitor (CD47

blocking mAb B6H12) using fixed cells in the LSC-based assay to our biochemical assay in a 1536 well plate format (**Fig 5C**) (as described in [33]). The IC50 of B6H12 in the LSC assay was 6 nM and 1 nM for 384 and 1536 well respectively, compared to 10 nM for the biochemical assay. The shift in potency for the LSC assay in 384 (live cells) to 1536 (fixed cells) wells likely reflects the same shift due to fixation as seen in **Fig 5A**. Both LSC assay formats compare well to our biochemical screening assay.

## Cell-based analysis of SM inhibitors previously identified in a CD47-SIPRα qHTS

In our recent publication[33], we described the development of a series of biochemical assays for the discovery of SIRPα-CD47 inhibiting SMs using qHTS. In order to characterize validated biochemical hits and optimize them toward lead molecules, it is important to demonstrate their activity in cell-based models. For this purpose, we used the LSC assay to measure SM antagonism of SIRPα binding to CD47 naturally expressed on tumor cells and compared the activity of these qHTS active SMs to the activity observed in the biochemical assay. **Fig 6A** shows a representative set of chemical analogs based on the parent compound identified from qHTS (NCGC00138783). These compounds illustrate a typical attempt to establish a structure activity relationship to enable optimization of potency and drug-like properties via medicinal chemistry. Concentration-response curves for the SMs in the biochemical and cell-based

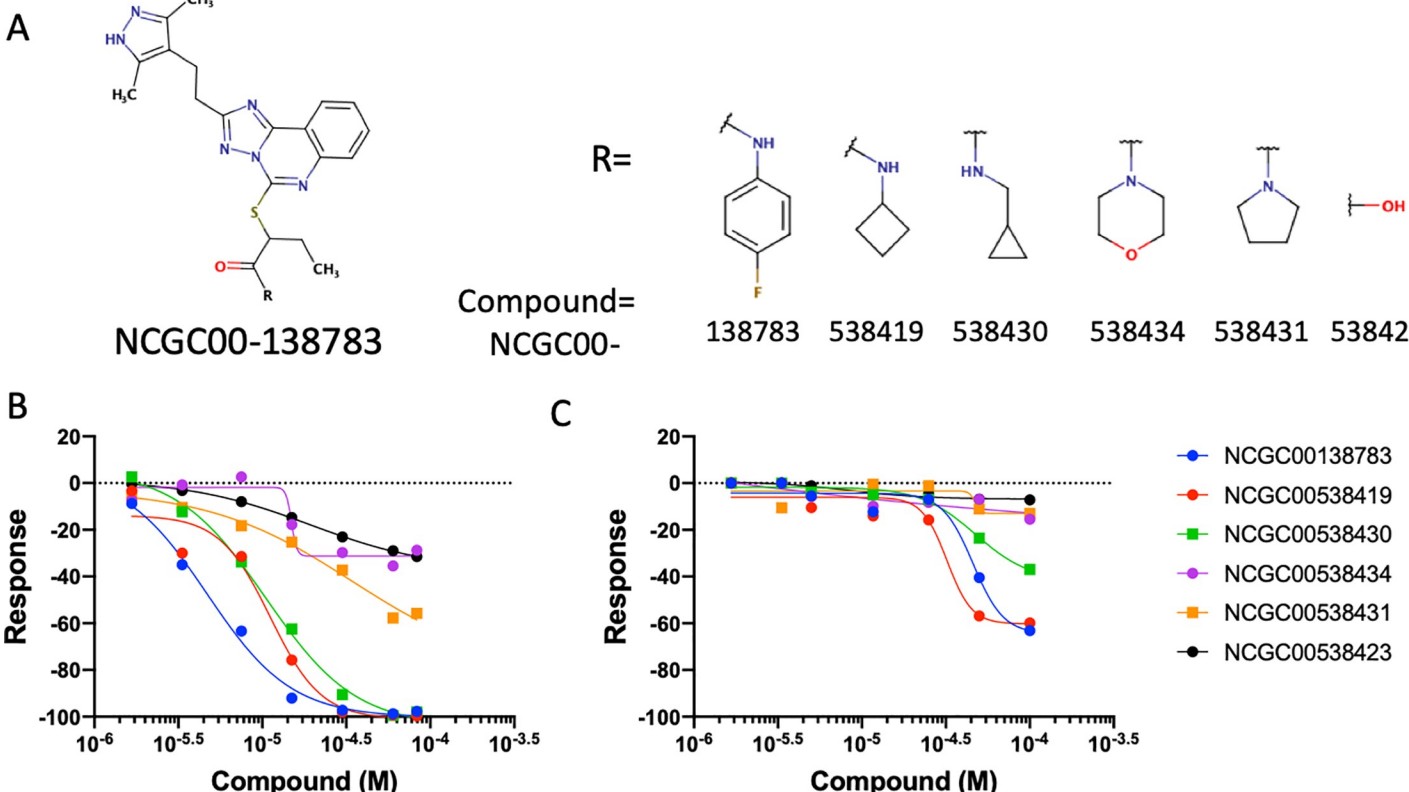

**Fig 6. Comparing small molecule activity profiles using the LSC assay and the biochemical assay.** (A) The structures of the compounds compared to the parent screening active compound (NCGC00138783). (B) Biochemical (ALPHAScreen) SIRPα-CD47 antagonism activity of NCGC00138783 and the structurally related analogs shown in A were compared to (C) antagonism activity of SIRPα binding to CD47 expressed on live Jurkat cells in the LSC assay. Normalized activity calculated with neutral control (CD47 + SIRPα, no inhibitor) as 0% and negative control (CD47—SIRPα, no inhibitor) as -100% activity. Assay activity with tested agents was then compared and normalized to these controls.

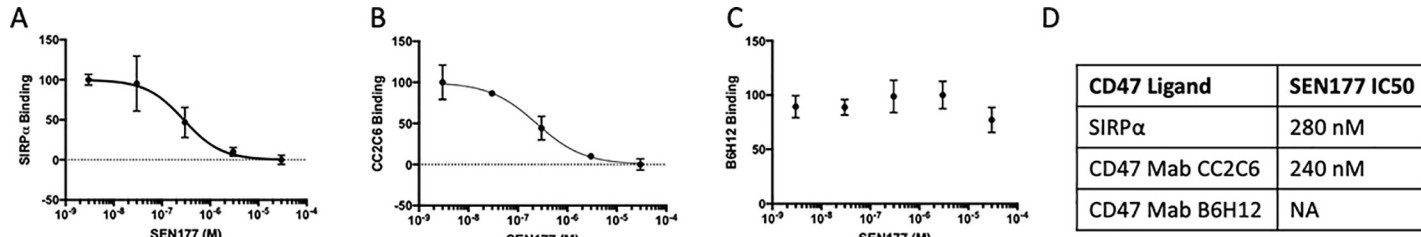

**Fig 7. The LSC assay can be used to measure SIRPα binding sensitivity to CD47 pyroglutamate modification.** (A) A2058 melanoma cells were treated with a range of concentrations of SEN177 for 48 hours and SIRPα binding quantified using the LSC assay. (B) Anti-CD47 clone CC2C6 or (C) clone B6H12 reactivity for the same samples as in panel A. The IC50 value is indicated where measurable. Data represents n = 3 replicates measurements for 3 separate experiments.

assays are presented in **Fig 6B and 6C**, respectively. As shown in the panels, the compounds displayed a range of potencies in both assays, with greater inhibitory activity evident in the biochemical assay. Nonetheless, cell-based activity was confirmed for the three most potent compounds.

## LSC assay confirmed that CD47 post-translational modification is required for SIRPα-CD47 binding

Consistent with interactions inferred from the **SIRPα-CD47** co-crystal structure[42,43], SIRPα-CD47 binding critically depends on a CD47 post-translational modification of its N-terminal glutamine to a pyroglutamate[36]. This modification in tumor cells depends on the activity of glutaminyl cyclase (QPCTL) and can be inhibited pharmacologically by the pan-glutaminyl cyclase inhibitory molecule SEN177[45]. We used the LCS assay to validate and extend these findings by treating A2058 melanoma cells with SEN177 to inhibit QPCTL and block the N-terminal pyroglutamate modification of CD47 (**Fig 7**). As in [36], we observed a reduction in SIRPα binding after 48 h treatment with SEN177 (**Fig 7A**). This was also associated with a similar reduction in the reactivity of clone CC2C6 antibody, whose epitope is known to be sensitive to the N-terminal pyroglutamate modification[39] (**Fig 7B**). Total CD47 levels were not significantly affected as judged by a lack of SEN177 treatment-associated changes in cell surface binding of the CD47 antibody clone B6H12, which is insensitive to N-terminal pyroglutamate modification[36] (**Fig 7C**). In addition to validating the impact of the glutaminyl cyclase inhibitor first reported by Logtenberg et al., we determined the IC50 of SEN177 inhibition of SIRPα binding to be 280 nM. This indicates a lower potency than the IC50 reported for inhibition of glutaminyl cyclase activity in a biochemical assay (20 nM[46]), but is consistent with the efficacy of the concentration (10 μM) employed in cells by Logtenberg et al.

## Discussion

Maximizing the success of immunotherapies in oncology care will depend on the identification of additional immune checkpoint molecules that are co-opted by tumors, the mitigation of immune related side-effects of those currently targeted by biologics in the clinic and the use of more effective combinations of immunotherapies and immune adjuvants. Small molecules could have an important role in achieving these goals. They also have uses as chemical probes to validate new anti-tumor immune checkpoints and as new therapeutic agents with lower toxicity and/or oral delivery to replace biologics for established targets or directed against novel targets. Their therapeutic also could enable better combinations with other immuno-oncology or conventional cancer therapies. The ability to monitor small molecule modulation of cell surface immune checkpoint interactions in their native environment is vital to advance these

objectives. Past studies have used radiolabeled ligand binding assays and flow cytometry to monitor cell-based receptor-ligand or receptor-counter receptor interactions. However, these techniques have important drawbacks that diminish their applicability to HTS-based drug discovery campaigns, such as limits on radioactivity usage and low throughput.

In this report, we describe the development and optimization of a cell-based binding assay utilizing a LSC platform[35,47]. The assay was designed to support the in vitro characterization and validation of SMs that disrupt the SIRPα-CD47 protein-protein interaction in a cell-free biochemical assay (see Miller et al. 2019). The LSC assay is a critical component of our SM drug-development program to identify novel agents that activate patients' innate immune system for cancer immunotherapy. To our knowledge, this is the only cell-based, ligand-binding assay to model a protein-protein interaction reported to deploy the Mirrorball LSC platform [48]. In particular, this assay provides a homogeneous platform to model a cell-cell interaction mediated by CD47 and SIRPα using a simplified single cell configuration.

The final assay that we developed has a low volume, high capacity homogeneous format that, in combination with our biochemical assay, will enable the screening thousands of compounds to identify active SMs, develop structure activity relationships (SAR) and support the optimization of hits to leads by a typical iterative medicinal chemistry campaign. We demonstrated that the assay is specific, concentration dependent and works well for the two major human expressed variants of SIRPα (V1 and V2). Results with our LSC assay parallel those obtained with our biochemical assay as well as the previously published studies by others. Notably, we confirmed the recent findings that treatment of cells with the glutaminyl cyclase inhibitor, SEN177, blocked SIRPa binding (**Fig 7** and Logtenberg et al.). Furthermore, the LSC assay is capable of quantitatively measuring the SIRPα-CD47 interaction in both live and fixed tumor cells; as a practical matter, this significantly reduces the cell culture burden and allows for the use of stable, fixed cell standards to control for inter- and intra-assay variations. Although we have focused here on SIRPα-CD47, the LSC-based cellular receptor ligand binding assay is also applicable to other cell surface receptor-ligand or receptor-counter-receptor complexes (including other immune checkpoints), especially in situations where producing the recombinant receptor is problematic or where its activity is significantly altered outside of the native cellular environment.

Because detection of the binding signal in the LSC assay was targeted specifically to cells marked by a nuclear stain and limited to the cell-containing focal plane by confocal optics, there was no need to remove non-binding reagents by including washing steps. Under these conditions, SIRPα is presented to CD47 on the cells as a multimeric reagent bound to tetrameric NAV. This presumably contributed to a stronger binding affinity of SIRPα and CD47 relative to measurements in a standard flow cytometry format where SIRPα is first presented as a monomer and non-bound reagent is removed by washing before exposure to SAV. While the higher avidity of SIRPα-CD47 interaction in the LSC assay would make it more difficult to observe the inhibitory effect of compounds having substantially lower potency, we were able to detect specific inhibition by a number of compounds that were first identified in our qHTS of libraries containing tens of thousands of SMs. Thus, we were able to validate active SMs from our cell-free screen and follow up by testing structurally related analogs to begin to determine SAR for the compound series. Because these are the first SM inhibitors of the SIRPα-CD47 interaction to be described, there were no comparable agents to use as positive controls in the study. Therefore, we used the well-characterized and very potent (nM) antibodies that are known to block the SIRPα-CD47 interaction. The goal of our work is to optimize small molecules discovered through unbiased qHTS into molecular probes with sufficient potency and selectivity for in vivo evaluation and potential therapeutic development. This will require optimized molecules with potency approaching that of the blocking antibodies; the levels of

potency and efficacy will be defined in biological assays like phagocytosis and in vivo tumor models. We view the LSC assay as a valuable tool to demonstrate that molecules identified in the initial discovery phase have activity in a cellular environment and prioritize them for optimization to the potency required for more complex biological evaluation.

The binding of recombinant CD47 with SIRPα was previously determined using SPR with immobilized CD47. Here we titrated soluble CD47 to measure its ability to bind soluble SIRPα and antagonize the interaction of the latter with cell-expressed CD47. Using this system, we demonstrated similar affinities when CD47 and SIRPα were permitted to interact in solution as compared to their interaction when CD47 was immobilized for an SPR assay (**Fig 4B and 4C**).

The ability of SIRPα to bind CD47 on paraformaldehyde-fixed cells raises the possibility that this reagent also might bind specifically to CD47 in formalin-fixed, paraffin-embedded tissue specimens. If this were borne out, then such an assay could be used to identify tumors with high SIRPα-CD47 binding capacity and therefore more likely to be responsive to immunotherapy targeting this interaction. The detection of high affinity SIRPα binding on tumor cells may prove to be a better indicator of response to therapy than measures of CD47 mRNA or protein expression.

## Supporting information

**S1 Material.**
(PDF)

## Acknowledgments

We gratefully acknowledge the contributions of the NCATS automation group (Jamie Travers, Carleen Klump-Thomas, Sam Michael), NCATS compound management group (Paul Shinn, Mischa Itkin, Zina Itkin, Crystal McKnight, Kamaria Butler), NCATS informatics (Yuhong Wang, Noel Southall and Bryan Queme), NHLBI core facility, the Protein Expression Laboratory (PEL) and Kirill Gorshkov (NCATS Division of Developmental Therapeutics) for helpful discussions in optimizing and troubleshooting studies using the Mirrorball instrument.

## Author Contributions

**Conceptualization:** Teresa L. Burgess, Joshua D. Amason, Jeffrey S. Rubin, David D. Roberts, Catherine L. Farrell, James Inglese, Thomas W. Miller.

**Data curation:** Joshua D. Amason, Damien Y. Duveau, Laurence Lamy, Thomas W. Miller.

**Formal analysis:** Teresa L. Burgess, Joshua D. Amason, Jeffrey S. Rubin, Damien Y. Duveau, Laurence Lamy, Thomas W. Miller.

**Funding acquisition:** Teresa L. Burgess, Jeffrey S. Rubin, David D. Roberts, Catherine L. Farrell, Thomas W. Miller.

**Investigation:** Joshua D. Amason, Damien Y. Duveau, Laurence Lamy, Thomas W. Miller.

**Methodology:** Joshua D. Amason, Jeffrey S. Rubin, Damien Y. Duveau, Laurence Lamy, David D. Roberts, Catherine L. Farrell, James Inglese, Thomas W. Miller.

**Project administration:** Teresa L. Burgess, Catherine L. Farrell, Thomas W. Miller.

**Resources:** David D. Roberts, Catherine L. Farrell, James Inglese, Craig J. Thomas, Thomas W. Miller.

**Software:** Thomas W. Miller.

**Supervision:** Teresa L. Burgess, Craig J. Thomas, Thomas W. Miller.

**Validation:** Jeffrey S. Rubin, Thomas W. Miller.

**Visualization:** James Inglese, Thomas W. Miller.

**Writing – original draft:** Teresa L. Burgess, Joshua D. Amason, Jeffrey S. Rubin, Damien Y. Duveau, Thomas W. Miller.

**Writing – review & editing:** Teresa L. Burgess, Jeffrey S. Rubin, Damien Y. Duveau, Laurence Lamy, David D. Roberts, Catherine L. Farrell, James Inglese, Craig J. Thomas, Thomas W. Miller.

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
