## [Decision Letter · Decision Letter 0]

27 Jan 2020

PONE-D-19-33480

A Homogeneous SIRPα-CD47 Cell-Based, Ligand-Binding Assay:  Utility for Small Molecule Drug Development in Immuno-oncology.

PLOS ONE

Dear Dr. Miller,

Thank you for submitting your manuscript to PLOS ONE. After careful consideration, we feel that it has merit but does not fully meet PLOS ONE’s publication criteria as it currently stands. Therefore, we invite you to submit a revised version of the manuscript that addresses the points raised during the review process.

We would appreciate receiving your revised manuscript by Mar 12 2020 11:59PM. To enhance the reproducibility of your results, we recommend that if applicable you deposit your laboratory protocols in protocols.io, where a protocol can be assigned its own identifier (DOI) such that it can be cited independently in the future. For instructions see: http://journals.plos.org/plosone/s/submission-guidelines#loc-laboratory-protocols

We look forward to receiving your revised manuscript.

Kind regards,

Irina V. Lebedeva, Ph.D.

Academic Editor

PLOS ONE

"391 This work was supported in part by the Avon Foundation for Women (02-2015-045, 02-2014-051, 02-2015-095, 02-2017-034; TWM, TLB, CLF), the National Cancer Institute (1U01CA218259-01A1; TWM, TLB, CLF, JSR) the Intramural Research Programs of NCATS (Project 1ZIATR000053-03, JI) and NCI (Project ZIASC009172, DDR), and Cooperative Research and Development Agreement 02921 with the National Institutes of Health. The funders had no role in study design, data collection and analysis, decision to publish, or preparation of the manuscript."

"This work was supported in part by the Avon Foundation for Women (02-2015-045, 02-2014-051, 02-2015-095, 02-2017-034; TWM, TLB, CLF), the National Cancer Institute (1U01CA218259-01A1; TWM, TLB, CLF, JSR) the Intramural Research Programs of NCATS (Project 1ZIATR000053-03, JI) and NCI (Project ZIASC009172, DDR)."

4. Thank you for stating the following in the Financial Disclosure section:

"This work was supported in part by the Avon Foundation for Women (02-2015-045, 02-2014-051, 02-2015-095, 02-2017-034; TWM, TLB, CLF), the National Cancer Institute (1U01CA218259-01A1; TWM, TLB, CLF, JSR) the Intramural Research Programs of NCATS (Project 1ZIATR000053-03, JI) and NCI (Project ZIASC009172, DDR)."

We note that one or more of the authors are employed by a commercial company: Paradigm Shift Therapeutics LLC.

B) Please also provide an updated Competing Interests Statement declaring this commercial affiliation along with any other relevant declarations relating to employment, consultancy, patents, products in development, or marketed products, etc.  

Reviewers' comments:

Reviewer's Responses to Questions

**Comments to the Author**

1. Is the manuscript technically sound, and do the data support the conclusions?

Reviewer #1: Yes

Reviewer #2: Yes

2. Has the statistical analysis been performed appropriately and rigorously? 

Reviewer #1: Yes

Reviewer #2: Yes

3. Have the authors made all data underlying the findings in their manuscript fully available?

Reviewer #1: Yes

Reviewer #2: Yes

4. Is the manuscript presented in an intelligible fashion and written in standard English?

Reviewer #1: Yes

Reviewer #2: Yes

5. Review Comments to the Author

Reviewer #1: The paper by Burgess et al describes the development of a homogeneous cellular assay to detect the CD47/SIRPalpha interaction as a potential target for small molecule immunotherapy. The topic is timely and important. The paper is very well organized and clearly written. The data support their conclusion that the assay is capable to detect small molecule disruptors of CD47/SIRPalpha. I have only a few minor points for clarification.

Abstract, line 9. Sentence on pyroglutamate is unclear – please check grammar.

Page 5 what was the concentration of SIRPalpha-biotin used?

Page 16, line 8. Should “transferase” read “cyclase”?

Figure 6. Y axis appears normalized. If so, how were top and bottom defined?

Reviewer #2: Manuscript focused on development of cell based assay to screen/characterize inhibitors of CD47/SIRP1a. Characterization of assay and comparison to prior method is detailed and rigorous.

Suggested improvements as follows:

1. Data characterizing molecules is limited to the CD47/SIRP1a blocking assays - either the novel cell based assay described here or the previously described protein/protein based assay. A bridging experiment demonstrating that the molecules identified from the screening actually impact CD47 biology and response in a primary cell based assay (for example a phagocytosis assay) would increase relevance of the assay to identify novel inhibitors or to guide molecule optimization.

2. The small molecules appear to work relatively inferior to the benchmark reference CD47 antibodies in the cell based assay (compare Figure 4A/B versus Figure 6C). Some recognition/explanation of this in the discussion is warranted. Perhaps the assay can be used to screen for modifications of the small molecules or for others that can reach the activity of a blocking antibody ?

3. Regarding the last paragraph of the Discussion - have the authors tried to detect SIRPa reagent binding to formalin fixed tumor cells ? Inclusion of such data would justify inclusion of the paragraph and would seem relatively straightforward to test.

6. PLOS authors have the option to publish the peer review history of their article (what does this mean?). If published, this will include your full peer review and any attached files.

Reviewer #1: No

Reviewer #2: No

---

## [Author Response · Author response to Decision Letter 0]

20 Feb 2020

Reviewer #1: 

The paper by Burgess et al describes the development of a homogeneous cellular assay to detect the CD47/SIRPalpha interaction as a potential target for small molecule immunotherapy. The topic is timely and important. The paper is very well organized and clearly written. The data support their conclusion that the assay is capable to detect small molecule disruptors of CD47/SIRPalpha. I have only a few minor points for clarification.

1. Abstract, line 9. Sentence on pyroglutamate is unclear – please check grammar.

Response: Thank you for taking the time to carefully review our manuscript. We appreciate that you found the work to be well organized and clearly written on a timely topic. We have addressed the minor typographical errors in the text. With regard to comment #1, we have clarified the grammar for the abstract sentence describing the detection of pyroglutamate sensitive binding with our assay. “We also utilized the LSC assay to confirm published studies showing that the inhibition of amino-terminal pyroglutamate formation on CD47 using the glutaminyl cyclase inhibitor SEN177 disrupts SIRPα binding.”

2. Page 5 what was the concentration of SIRPalpha-biotin used?

Response: We have indicated the concentrations as in Figure 1 legend (0.1, 0.3, 1.0 �M

Page 16, line 8. Should “transferase” read “cyclase”?

Response: Yes, you are correct. The text has been updated to read “cyclase”.

Figure 6. Y axis appears normalized. If so, how were top and bottom defined?

Response: Yes, the Y axes are normalized in Figure 6. The detailed normalization procedure is now included in the figure legend.

Reviewer #2: Manuscript focused on development of cell based assay to screen/characterize inhibitors of CD47/SIRP1a. Characterization of assay and comparison to prior method is detailed and rigorous. 

1. Data characterizing molecules is limited to the CD47/SIRP1a blocking assays - either the novel cell based assay described here or the previously described protein/protein based assay. A bridging experiment demonstrating that the molecules identified from the screening actually impact CD47 biology and response in a primary cell based assay (for example a phagocytosis assay) would increase relevance of the assay to identify novel inhibitors or to guide molecule optimization.

Response: Thank you for taking the time to carefully review our manuscript. We appreciate that you found the work to be detailed and rigorous. Regarding the first comment, we agree that the relevance of the small molecule SIRPa-CD47 inhibitors described in the manuscript would be enhanced by further mechanistic data showing that they can impact the biology of SIRPa-CD47 signaling. We present the molecules as tool compounds capable of disrupting CD47-SIRPa binding in biochemical assays and our novel cell-based ligand binding assay. However, as you point out below, these molecules are far from the potency of CD47 or SIRPa specific monoclonal antibodies and are not expected to impact biological responses without further optimization. In our macrophage-tumor cell co-culture phagocytosis assay, the anti-CD47 antibody B6H12 potentiates phagocytosis by 2-fold with an EC50 of 1.7 nM. Based on their activity in the LSC assay, the small molecules described here are not sufficiently potent/soluble to show significant activity in the phagocytosis assay. As described in the text, we view the LSC assay as an intermediate assay between the biochemical assay and other in vitro assays that can assess the potency of a molecule in a cellular context and guide its optimization for functional in vitro activity. 

2. The small molecules appear to work relatively inferior to the benchmark reference CD47 antibodies in the cell based assay (compare Figure 4A/B versus Figure 6C). Some recognition/explanation of this in the discussion is warranted. Perhaps the assay can be used to screen for modifications of the small molecules or for others that can reach the activity of a blocking antibody?

Response: The molecules described in the manuscript are early screening hits that we used as tool compounds to validate our CD47-SIRPa assays and provide a basis for optimization into more potent chemical probes. As expected at this stage, these molecules are much less potent than the anti-CD47 antibodies used as positive controls for the LSC assay and they should not be viewed as equivalent tools. As described above, our goal in creating and detailing the LSC assay in the manuscript is to provide a means to compare the activity of small molecules (and other modalities) during their iterative medchem optimization in a context where they can be evaluated alongside benchmark blocking antibodies for a variety of tumor cell types. This has been further described and clarified in the discussion to better convey the relevance of the present study. “Because these are the first SM inhibitors of the SIRPα-CD47 interaction to be described, there were no comparable agents to use as positive controls in the study. Therefore, we used the well-characterized and very potent (nM) antibodies that are known to block the SIRPα-CD47 interaction. The goal of our work is to optimize small molecules discovered through unbiased qHTS into molecular probes with sufficient potency and selectivity for in vivo evaluation and potential therapeutic development. This will require optimized molecules with potency approaching that of the blocking antibodies; the levels of potency and efficacy will be defined in biological assays like phagocytosis and in vivo tumor models. We view the LSC assay as a valuable tool to demonstrate that molecules identified in the initial discovery phase have activity in a cellular environment and prioritize them for optimization to the potency required for more complex biological evaluation.”

3. Regarding the last paragraph of the Discussion - have the authors tried to detect SIRPa reagent binding to formalin fixed tumor cells? Inclusion of such data would justify inclusion of the paragraph and would seem relatively straightforward to test.

Response: Our studies used a solution of 4% paraformaldehyde to fix tumor cells prior to analysis. This is similar to but not precisely formalin fixation (typically a 10% neutral buffered solution of 37% formaldehyde). The manuscript discussion is updated to correctly refer to the tumor cells as being “paraformaldehyde fixed” while maintaining the speculation that the methodology may be applicable to formalin-fixed tissue samples. While we have encouraging results in pilot experiments evaluating SIRPa binding in fixed tumor samples, these data are not mature enough for publication. We believe that their inclusion or further elaboration would detract from the main purpose of this manuscript—to describe the LSC. If the reviewer disagrees, we will remove the speculative paragraph from the discussion.

---

## [Decision Letter · Decision Letter 1]

9 Mar 2020

A Homogeneous SIRPα-CD47 Cell-Based, Ligand-Binding Assay:  Utility for Small Molecule Drug Development in Immuno-oncology.

PONE-D-19-33480R1

Dear Dr. Miller,

We are pleased to inform you that your manuscript has been judged scientifically suitable for publication and will be formally accepted for publication once it complies with all outstanding technical requirements.

With kind regards,

Irina V. Lebedeva, Ph.D.

Academic Editor

PLOS ONE

Additional Editor Comments (optional):

Reviewers' comments:

Reviewer's Responses to Questions

**Comments to the Author**

1. If the authors have adequately addressed your comments raised in a previous round of review and you feel that this manuscript is now acceptable for publication, you may indicate that here to bypass the “Comments to the Author” section, enter your conflict of interest statement in the “Confidential to Editor” section, and submit your "Accept" recommendation.

Reviewer #1: All comments have been addressed

Reviewer #2: All comments have been addressed

2. Is the manuscript technically sound, and do the data support the conclusions?

Reviewer #1: Yes

Reviewer #2: Yes

3. Has the statistical analysis been performed appropriately and rigorously? 

Reviewer #1: Yes

Reviewer #2: Yes

4. Have the authors made all data underlying the findings in their manuscript fully available?

Reviewer #1: Yes

Reviewer #2: Yes

5. Is the manuscript presented in an intelligible fashion and written in standard English?

Reviewer #1: Yes

Reviewer #2: Yes

6. Review Comments to the Author

Reviewer #1: (No Response)

Reviewer #2: Manuscript well written and concerns highlighted in initial submission have been addressed in the resubmission.

7. PLOS authors have the option to publish the peer review history of their article (what does this mean?). If published, this will include your full peer review and any attached files.

Reviewer #1: No

Reviewer #2: No

---

## [Editor Report · Acceptance letter]

18 Mar 2020

PONE-D-19-33480R1 

A Homogeneous SIRPα-CD47 Cell-Based, Ligand-Binding Assay:  Utility for Small Molecule Drug Development in Immuno-oncology. 

Dear Dr. Miller:

I am pleased to inform you that your manuscript has been deemed suitable for publication in PLOS ONE. Congratulations! Your manuscript is now with our production department. 

With kind regards,

on behalf of

Dr. Irina V. Lebedeva 

Academic Editor

PLOS ONE